# Stage-Specific Transcriptomic Insights into Seed Germination and Early Development in *Camellia oleifera* Abel.

**DOI:** 10.3390/plants14152283

**Published:** 2025-07-24

**Authors:** Zhen Zhang, Caixia Liu, Ying Zhang, Zhilong He, Longsheng Chen, Chengfeng Xun, Yushen Ma, Xiaokang Yuan, Yanming Xu, Rui Wang

**Affiliations:** 1Hunan Academy of Forestry, Changsha 410004, China; hfazz@hnlky.cn (Z.Z.); lcaixia1989@hnlky.cn (C.L.); zhangying@hnlky.cn (Y.Z.); hezhilong2000@hnlky.cn (Z.H.); clongsheng@163.com (L.C.); xunchengfeng24@163.com (C.X.); mys9204@163.com (Y.M.); 2National Engineering Research Center for Oil Tea Camellia, Changsha 410004, China; 3Yuelushan Laboratory, Changsha 410004, China; 4State Key Laboratory of Woody Oil Resources Utilization, Changsha 410004, China; 5Hunan Key Laboratory of Meteorological Disaster Prevention and Reduction, Hunan Research Institute of Meteorological Sciences, Changsha 410118, China; yxknuist@126.com

**Keywords:** *Camellia oleifera*, seed germination, transcriptome sequencing, differentially expressed genes

## Abstract

Seed germination is a critical phase in the plant lifecycle of *Camellia oleifera* (oil tea), directly influencing seedling establishment and crop reproduction. In this study, we examined transcriptomic and physiological changes across five defined germination stages (G0–G4), from radicle dormancy to cotyledon emergence. Using RNA sequencing (RNA-seq), we assembled 169,652 unigenes and identified differentially expressed genes (DEGs) at each stage compared to G0, increasing from 1708 in G1 to 10,250 in G4. Functional enrichment analysis revealed upregulation of genes associated with cell wall organization, glucan metabolism, and Photosystem II assembly. Key genes involved in cell wall remodeling, including cellulose synthase (CESA), phenylalanine ammonia-lyase (PAL), 4-coumarate-CoA ligase (4CL), caffeoyl-CoA O-methyltransferase (COMT), and peroxidase (POD) showed progressive activation during germination. A Kyoto Encyclopedia of Genes and Genomes (KEGG) pathway analysis revealed dynamic regulation of phenylpropanoid and flavonoid biosynthesis, photosynthesis, carbohydrate metabolism, and hormone signaling pathways. Transcription factors such as indole-3-acetic acid (IAA), ABA-responsive element binding factor (ABF), and basic helix–loop–helix (bHLH) were upregulated, suggesting hormone-mediated regulation of dormancy release and seedling development. Physiologically, cytokinin (CTK) and IAA levels peaked in G4, antioxidant enzyme activities were highest in G2, and starch content increased toward later stages. These findings provide new insights into the molecular mechanisms underlying seed germination in *C. oleifera* and identify candidate genes relevant to rootstock breeding and nursery propagation.

## 1. Introduction

Seed germination is the fundamental and most critical phase of plant life. It plays an essential role in shaping seedling growth quality and determining the plant’s future vitality. Seed development involves complex physiological and metabolic processes that are controlled by intricate molecular mechanisms, including hormone signaling, stress responses, and metabolic pathway regulation. In model and crop species such as *Arabidopsis thaliana*, *Oryza sativa*, and *Zea mays*, germination has been extensively studied, revealing the importance of pathways related to cell wall remodeling, nutrient mobilization, and reactive oxygen species (ROS) detoxification [1,2,3]. However, there remains limited information on the molecular mechanisms governing seed germination in *C. oleifera*, an economically important oil-producing species [4]. This study applies transcriptomic and physiological analyses to elucidate the regulatory networks controlling seed germination in *C. oleifera.* By focusing on well-characterized pathways such as phenylpropanoid biosynthesis, hormone signaling, and antioxidant defense, we aim to provide new insights that can support improved breeding and cultivation strategies for this species.

*Camellia oleifera* Abel. is an evergreen woody oilseed tree species native to China, and it is recognized as one of the world’s four major woody oil crops, alongside the oil palm (*Elaeis guineensis*), olive (*Olea europaea*), and coconut (*Cocos nucifera*) [5]. Its seeds yield high-quality edible oil rich in unsaturated fatty acids, tocopherols, and polyphenols, which are widely praised for both nutritional and medicinal value. In addition to its contribution to food security, the development of the *C. oleifera* industry has become a national strategic priority in China. It plays a vital role in promoting rural revitalization, enhancing ecological sustainability, improving regional economic structures, and increasing incomes in forest-dependent communities [6]. The propagation and productivity of *C. oleifera* heavily depend on seed-derived rootstocks, making seed germination and early seedling development critical stages for nursery success, grafting efficiency, and large-scale cultivation [7,8]. Currently, the Chinese government is actively promoting a national three-year action plan to accelerate *C. oleifera* industry development, which places urgent demands on high-quality seedling and rootstock production. Grafting with seedling rootstocks is the primary method of propagation, and the quality of rootstock seedlings derived from germinated seeds directly influences grafting success and subsequent plantation performance. As such, a detailed understanding of seed germination and early development at the molecular level is essential for optimizing rootstock production, improving grafting efficiency, and ensuring the cultivation of elite *C. oleifera* cultivars with high oil yield and quality. *C. oleifera* is cultivated primarily for its oil-rich seeds, which accumulate substantial quantities of triacylglycerols during seed development. These lipids serve as the main energy source during germination and are essential for fueling early seedling growth. While the biosynthesis and composition of seed oils in *C. oleifera* have been relatively well characterized [9,10,11,12], the mobilization and metabolic regulation of these reserves during germination remain poorly understood. Elucidating how lipid reserves are utilized during germination is crucial for understanding the energy metabolism underlying seedling establishment. This knowledge may further contribute to improving rootstock vigor and optimizing oilseed propagation strategies, both of which are central to the sustainable development of the *C. oleifera* industry.

Seed germination marks a crucial developmental transition from embryonic dormancy to active growth and is a key determinant of seedling vigor. This process is regulated by a complex interplay of endogenous genetic programs and environmental cues, including temperature, light, water availability, and nutrient conditions [13,14,15]. During germination, numerous genes are differentially expressed to coordinate metabolic reactivation, hormonal signaling, oxidative stress response, cell wall modification, and starch degradation. Transcriptome analyses in model and crop species such as *Arabidopsis thaliana*, rice (*Oryza sativa*), maize (*Zea mays*), adzuki bean (*Vigna angularis*), and *Calanthe tsoongiana* have uncovered critical gene networks and regulatory pathways that underpin these developmental processes [16,17,18,19]. Despite its agricultural and economic importance, the molecular mechanisms underlying seed germination and early seedling development in *C. oleifera* remain poorly understood. Most existing research has focused on physiological aspects such as dormancy-breaking treatments, temperature sensitivity, hormonal regulation, and nutrient supply, while the transcriptional dynamics that govern this critical developmental stage have received limited attention [20]. With the advancement of next-generation sequencing technologies, transcriptome analysis has become a powerful tool for exploring gene expression landscapes in non-model species. High-throughput RNA sequencing (RNA-seq) enables the identification of differentially expressed genes (DEGs), functional annotation, and metabolic pathway enrichment, providing a comprehensive view of transcriptional dynamics during developmental transitions [17,18].

This study aims to investigate the molecular mechanisms underlying the germination and early growth of *C. oleifera* seeds across five key developmental stages, from the onset of radicle emergence to cotyledon development. Specifically, the study will examine stages such as the radicle not yet breaking through the seed coat, radicle just breaking through the seed coat, radicle extending to 2 cm, radicle extending to 5 cm, and cotyledon emergence. We hypothesize that specific molecular processes govern these stages. The objectives of the study were: (1) to profile stage-specific gene expression patterns and changes in physiological and biochemical indicators, (2) to identify differentially expressed genes (DEGs) involved in hormone signaling, lignin biosynthesis, photosynthesis, and transcriptional regulation, and (3) to clarify the molecular networks controlling seed germination and early seedling growth. By conducting a comprehensive transcriptomic analysis of ‘Xianglin 27’ *C. oleifera* seeds, this study aims to explore the molecular regulatory mechanisms of seed germination and provide a solid foundation for future research on gene discovery related to seed growth and development. Additionally, the findings will contribute to optimizing rootstock varieties and cultivating high-quality *C. oleifera* seedlings, with significant applications in molecular breeding and the development of improved cultivars.

## 2. Results

### 2.1. Effects of Treatments on Phytohormones and Physiological–Biochemical Parameters

The physiological and biochemical characteristics of seeds under five germination stages (G0–G4) are summarized in Table 1. Significant differences were observed among germination stages in most measured parameters (*p* < 0.05). Overall, treatment G2 consistently promoted higher antioxidant activities and sugar accumulation, while G4 resulted in the highest hormone levels and starch content.

Both cytokinin (CTK) and IAA levels significantly increased across germination stages from G0 to G4. The highest CTK concentration was recorded in G4 (5.48 ± 0.10 μg/L) and was significantly higher than in G0 (4.14 ± 0.11 μg/L). Similarly, IAA content peaked in G4 (8.12 ± 0.18 μg/L), representing a notable increase over the control. No significant differences were found in photosynthetic pigments, chlorophyll a (Chl a), chlorophyll b (Chl b), or total chlorophyll (Chl a + b) among germination stages, although slight upward trends were observed, with G4 exhibiting the highest total chlorophyll content (0.022 ± 0.011 mg/g FW).

Soluble sugar and sucrose contents varied across germination stages. G2 showed the highest soluble sugar (75.73 ± 1.18 mg/g FW) and sucrose levels (130.43 ± 0.67 mg/g FW), while G0 had the lowest. Starch content was also highest in G4 (4.22 ± 0.05 mg/mg prot), indicating enhanced carbohydrate accumulation during this germination stage. Protein content was highest in G1 (10.78 ± 0.04 mg/g FW), significantly greater than in the control (G0), but declined in subsequent germination stages. The antioxidant enzyme activities superoxide dismutase (SOD) and POD were significantly enhanced by germination stages. G2 exhibited the highest SOD activity (27.48 ± 0.07 U/mg prot); POD peaked in the same germination stage (1.95 ± 0.02 ΔOD470/min/mg prot). CAT activity increased markedly in G3 (10.72 ± 0.01 μmol/min/mg prot), followed by G2 and G4. APX activity remained relatively stable among germination stages, with slight increases in G1 and G2. MDA and H_2_O_2_ contents reflected oxidative stress levels. MDA content increased significantly in G2 and G3 (0.71 ± 0.07 and 0.81 ± 0.01 nmol/mg prot, respectively), whereas H_2_O_2_ accumulation was highest in G1 (1.17 ± 0.01 μmol/mg prot) and significantly reduced in G4 (0.40 ± 0.01 μmol/mg prot). 

### 2.2. Transcriptome Sequencing and De Novo Assembly

Clean reads obtained from RNA sequencing were assembled using Trinity and further processed with TGICL for sequence clustering and redundancy removal. Germinated seeds of *C. oleifera* at different developmental stages were used for RNA extraction prior to sequencing. A total of 169,652 unigenes were successfully assembled, with a combined length of 148,782,822 bp. The unigenes had an average length of 877 bp and an N50 value of 1155 bp, indicating a high-quality assembly.

The length distribution of unigenes is presented in Appendix A. The majority of unigenes (40.77%) were within the range of 300–500 bp, followed by 32.88% between 501–1000 bp, 18.29% between 1001–2000 bp, and 8.06% exceeding 2000 bp in length.

### 2.3. Functional Annotation of Unigenes

A total of 169,652 unigenes from the *C. oleifera* transcriptome were functionally annotated by alignment against seven major public databases: NR, NT, Swiss-Prot, Pfam, GO, KOG, and KEGG (KO). Of these, 89,569 unigenes (52.79%) were successfully annotated in at least one database (Appendix A). The Pfam database contributed the highest number of annotations, with 100,023 unigenes (58.95%), followed by GO (57,706; 34.01%), NT (53,308; 31.42%), and Swiss-Prot (52,337; 30.84%). KEGG pathway annotation assigned 26,968 unigenes (15.89%), while the KOG and NR databases accounted for 19,812 (11.67%) and 19,512 (11.50%) unigenes, respectively. Notably, 2430 unigenes (1.43%) were consistently annotated across all seven databases, representing a core set of highly conserved and functionally characterized genes.

A total of 19,812 unigenes were annotated against the KOG (Eukaryotic Orthologous Groups) database and functionally classified into 25 categories (Appendix A). The largest group was category [J] “Translation, ribosomal structure and biogenesis,” accounting for approximately 16% of the annotated unigenes. This was followed by category [O] “Posttranslational modification, protein turnover, chaperones” (~15%) and category [R] “General function prediction only” (~11%). Other highly represented categories included [C] “Energy production and conversion,” [T] “Signal transduction mechanisms,” [U] “Intracellular trafficking, secretion, and vesicular transport,” [A] “RNA processing and modification,” [G] “Carbohydrate transport and metabolism,” and [E] “Amino acid transport and metabolism,” each comprising 5–7% of the total annotations. In contrast, categories such as [Y] “Nuclear structure,” [V] “Defense mechanisms,” [W] “Extracellular structures,” and [N] “Cell motility” were the least represented

A total of 57,706 unigenes were successfully annotated to the Gene Ontology (GO) database through sequence comparison and classified into three main functional categories: biological process, cellular component, and molecular function, encompassing a total of 56 subcategories (Appendix A). Within the biological process category, 26 subcategories were identified, with the most enriched being cellular processes and metabolic processes. The cellular component category included 20 subcategories, with cell and cell part being the most represented. In the molecular function category, 10 subcategories were defined, with a predominance of binding and catalytic activity.

A total of 26,968 unigenes were mapped to the KEGG database and assigned to three metabolic pathways (Appendix A), which were categorized into 5 primary classes and 19 secondary subcategories. Among these major classes, metabolism pathways were the most represented, encompassing 11 subcategories. Genetic information processing accounted for 4 subcategories, while organismal systems, cellular processes, and environmental information processing included 1, 1, and 2 subcategories, respectively. Of the 19 subcategories, the highest gene counts were found in pathways related to translation, carbohydrate metabolism, and folding, sorting, and degradation.

### 2.4. Differential Expressed Gene (DEG) Analysis

To investigate transcriptional changes across different germination stages, differential gene expression analysis was performed by comparing transcriptomic profiles of germination stages G1 to G4 against the control group G0. As presented in Table 2, the number of differentially expressed genes (DEGs) increased progressively with germination stage, indicating a progressively enhanced molecular response.

In the G1 vs. G0 comparison, 1708 DEGs were identified, including 1474 upregulated and 234 downregulated genes. In G2 vs. G0, the number of DEGs increased to 6505, comprising 5636 upregulated and 869 downregulated genes. A total of 8206 DEGs were detected in the G3 vs. G0 comparison, of which 7045 were upregulated and 1161 downregulated. The most substantial transcriptional shift was observed in G4 vs. G0, with 10,250 DEGs identified—8221 upregulated and 2029 downregulated—highlighting a strong activation of gene expression in response to germination stage progression.

### 2.5. GO Enrichment Analysis of Differentially Expressed Genes

Gene Ontology (GO) enrichment analysis was performed on the differentially expressed genes (DEGs) identified at each germination stage compared to the control (G0). As summarized in Appendix A, a variety of GO terms across the three main categories—biological process, cellular component, and molecular function—were significantly enriched. Within the cellular component category, a notable number of DEGs were associated with cell wall-related structures, including “cell wall” (31 to 81 genes), “external encapsulating structure” (39 to 103 genes), and “apoplast” (8 to 25 genes), showing an increasing trend in DEG counts from G1 to G4. Additionally, the “photosystem II oxygen evolving complex” was also enriched, with 7 to 17 genes across the stages.

In the biological process category, terms related to cell wall organization and modification showed significant enrichment. For example, “cell wall organization or biogenesis” included between 27 and 82 DEGs depending on the stage. Metabolic processes such as “cellular glucan metabolic process”, “polysaccharide metabolic process”, and “cellular carbohydrate metabolic process” also had increasing numbers of DEGs. The oxidation-reduction process was the most enriched biological process, with DEG counts rising from 137 at G1 to 724 at G4.

For the molecular function category, DEGs were enriched in functions such as “structural constituent of cell wall,” “heme binding,” and “oxidoreductase activity,” with the number of genes generally increasing with germination stage. Notably, “oxidoreductase activity” included 133 DEGs at G1, rising to 741 DEGs at G4, indicating a strong involvement of enzymes mediating redox reactions. Other enriched functions included “transferase activity,” “pectinesterase activity,” and “antioxidant activity,” underscoring the complexity of enzymatic functions engaged during germination stages.

### 2.6. KEGG Pathway Enrichment Analysis of DEGs

KEGG pathway enrichment analysis was conducted to identify the metabolic and signaling pathways significantly represented by the differentially expressed genes (DEGs) at each germination stage compared to the control (G0). As shown in Appendix A, several key pathways related to metabolism and photosynthesis were enriched, with DEG counts generally increasing as the germination stage progressed.

Notably, the phenylpropanoid biosynthesis pathway (ko00940) showed substantial enrichment, with the number of DEGs rising from 40 in G1 vs. G0 to 122 in G4 vs. G0. Within this pathway, a total of 11 DEGs were identified (Appendix A), including one beta-glucosidase (bglB), one caffeoyl-CoA O-methyltransferase (COMT; EC 2.1.1.104), two peroxidases (POD; EC 1.11.1.7), five phenylalanine ammo-nia-lyases (PAL; EC 4.3.1.24), and two 4-coumarate–CoA ligases (4CL). The expression levels of all 11 genes showed a consistent upward trend during *C. oleifera* seed gemination stages.

Photosynthesis-related pathways, such as photosynthesis—antenna proteins (ko00196) and photosynthesis (ko00195), were also enriched across all stages. DEG counts ranged from 16 to 41 genes in these pathways. In the photosynthesis—antenna proteins pathway (ko00196) of the KEGG database, a total of 16 common differentially expressed genes (DEGs) were identified (Appendix A), including 11 light-harvesting complex II chlorophyll a/b binding proteins (LHCB) and 5 light-harvesting complex I chlorophyll a/b binding proteins (LHCA). In the photosynthesis pathway (ko00195), 15 additional DEGs were identified, comprising 8 psb genes associated with Photosystem II (PSII) and 7 psa genes associated with Photosystem I (PSI). Expression levels of all 31 genes showed a consistent upregulation during the course of *C. oleifera* seed development, suggesting their positive regulatory roles in enhancing the photosynthetic capacity of developing seeds and supporting energy demands during maturation.

The flavonoid biosynthesis pathway (ko00941), another key secondary metabolism pathway, showed a steady increase from 19 DEGs in G1 to 40 in G4, suggesting enhanced flavonoid production. In the flavonoid biosynthesis pathway, eight differentially expressed genes (DEGs) were identified (Appendix A), including three chalcone synthase (CHS) genes, two anthocyanidin reductase (ANR) genes, one naringenin 3-dioxygenase (F3H), one leucoanthocyanidin dioxygenase (ANS), and one bifunctional dihydroflavonol 4-reductase/flavanone 4-reductase (DFR) gene. The expression levels of all eight genes were progressively upregulated during *C. oleifera* seed development.

Other significantly enriched pathways included starch and sucrose metabolism (ko00500) and plant hormone signal transduction (ko04075), both of which showed a notable increase in DEGs at later developmental stages. In the starch and sucrose metabolism pathway (Appendix A), four DEGs were identified: beta-fructofuranosidase (sacA), UDP-glucuronate 4-epimerase (GAE), sucrose synthase (SUS), and fructokinase (scrK). All four genes exhibited upregulated expression during seed development, indicating enhanced carbohydrate metabolism supporting energy accumulation in *C. oleifera* seeds. In the plant hormone signal transduction pathway (Appendix A), seven DEGs were identified. These included: two auxin-responsive protein IAA (IAA) genes and two auxin influx carrier (AUX1) genes involved in the auxin signaling pathway; one ABA-responsive element binding factor (ABF) in the abscisic acid (ABA) pathway; one cyclin D3 (CYCD3) gene in the brassinosteroid (BR) pathway; and one MYC2 transcription factor gene involved in jasmonic acid (JA) signaling. Additionally, a total of 27 transcription factor (TF)-related DEGs were identified (Appendix A), spanning major TF families including AP2/ERF-ERF, B3, bHLH, and MYB.

### 2.7. Validation of the Expression of DEGs by qRT-PCR

To validate the reliability of the transcriptomic data, nine DEGs associated with *C. oleifera* seed development (Appendix A) were selected for quantitative real-time PCR (qRT-PCR) analysis. Tubulin was used as the internal reference gene. The qRT-PCR results were largely consistent with the RNA-seq expression patterns (Appendix A).

## 3. Discussion

Seed germination is a highly regulated process involving complex physiological and biochemical changes that ensure the successful establishment of seedlings [21]. In this study, five different germination stages (G0 to G4) were applied to investigate their impact on various parameters, including phytohormones, carbohydrate metabolism, antioxidant enzyme activity, and oxidative stress markers. Significant differences were observed among the germination stages (*p* < 0.05), highlighting the germination stage-specific regulatory effects on seed development. Our findings revealed a continuous increase in both cytokinin (CTK) and indole-3-acetic acid (IAA) levels across germination stages from G0 to G4. Specifically, the highest CTK and IAA concentrations were recorded in G4 (5.48 ± 0.10 μg/L and 8.12 ± 0.18 μg/L, respectively). CTK plays a key role in promoting cell division and differentiation, particularly during early seedling development, likely contributing to the rapid growth of cotyledons and roots. Similarly, IAA, which promotes cell elongation, showed its peak concentration in G4, emphasizing its role in supporting root and shoot growth as the seedling progresses. The upregulation of these phytohormones demonstrates their integral role in regulating seed development [22].

Photosynthetic pigments, such as chlorophyll a (Chl a), chlorophyll b (Chl b), and total chlorophyll content, provide insight into the seed’s photosynthetic potential. While no significant differences were found between germination stages, a slight upward trend in chlorophyll content was observed, particularly in G4, which exhibited the highest total chlorophyll content (0.022 ± 0.011 mg/g FW). This increase in chlorophyll suggests that as seeds approach maturity, they become capable of initiating photosynthesis, which is vital for the subsequent growth of the seedling. The increase in chlorophyll at later stages of seed development aligns with findings in *Medicago sativa* [23], which reported that chlorophyll accumulation can enhance seed germination and seedling growth. The accumulation of soluble sugars, sucrose, and starch reflects the seed’s energy metabolism and storage capacity. Soluble sugars and sucrose levels were highest in G2 (75.73 ± 1.18 mg/g FW and 130.43 ± 0.67 mg/g FW), providing energy for the early stages of germination. Soluble sugars, in particular, are essential as a major energy source during the initial stages of germination [24]. Starch content, on the other hand, peaked in G4 (4.22 ± 0.05 mg/mg prot), indicating that starch, a primary carbon storage form in seeds, is increasingly utilized in later stages of seed development. This suggests that starch reserves are gradually broken down into soluble sugars to support the growth of the emerging seedling.

Protein content peaked in G1 (10.78 ± 0.04 mg/g FW), which is consistent with the rapid cell division and growth of the seedling during this early phase. However, protein levels declined in subsequent germination stages, which is likely due to the shift from protein synthesis to metabolic processes required for seed maturation and seedling establishment. The decrease in protein content after G1 could reflect the breakdown of stored proteins to provide amino acids and nitrogen for the developing seedling [25,26].

Antioxidant enzyme activities such as superoxide dismutase (SOD), peroxidase (POD), catalase (CAT), and ascorbate peroxidase (APX) play a crucial role in protecting seeds from oxidative damage [27]. SOD and POD activities were highest in G2, suggesting an enhanced ability to scavenge reactive oxygen species (ROS) during seedling emergence. The peak in CAT activity in G3 and the relatively stable APX activity across germination stages further indicate the gradual strengthening of the seed’s antioxidant defenses as it matures. In line with these findings, MDA and H_2_O_2_ levels, markers of oxidative stress, increased in G2 and G3, correlating with higher metabolic activity and ROS generation. However, H_2_O_2_ accumulation decreased significantly in G4, indicating improved oxidative stress management during the later stages of seedling development.

The coordinated regulation of phytohormones, carbohydrate metabolism, and antioxidant enzyme activities plays a pivotal role in seed development [28,29]. The increase in CTK and IAA levels promotes cell division, elongation, and differentiation, which are critical for successful seedling emergence. The dynamic changes in carbohydrate metabolism, particularly in soluble sugars, sucrose, and starch, provide the necessary energy for germination and seedling growth. The upregulation of antioxidant enzymes helps mitigate oxidative stress, ensuring the stability and viability of the seed during its development. Collectively, these physiological and biochemical changes contribute to the efficient transition from seed germination to seedling establishment. The study provides valuable insights into the complex physiological and biochemical processes that regulate seed development. The upregulation of phytohormones and metabolic pathways, along with the activation of antioxidant defenses, highlights the intricate coordination of biological processes that support seedling growth and establishment. Further research into the molecular mechanisms underlying these processes will enhance our understanding of seed biology and contribute to the development of strategies for improving crop yields and sustainability.

Transcriptome sequencing was conducted on *C. oleifera* seeds at different germination stages to explore key metabolic pathways and DEGs involved in germination. A total of 169,652 unigenes were assembled with a combined length of 148,782,822 bp, an average length of 877 bp, and an N50 of 1155 bp—indicating high assembly quality and completeness [30,31]. Most unigenes (40.77%) ranged from 300–500 bp, consistent with common functional elements like transcription factors and enzymes [32]. Only 8.06% exceeded 2000 bp, aligning with findings that longer transcripts are less abundant but may encode key regulatory genes [33]. The N50 value confirms the reliability of the assembly for functional annotation and pathway analysis, comparable to other plant transcriptome studies [34]. These results offer a valuable genetic resource for identifying candidate genes and understanding molecular mechanisms during *C. oleifera* seed germination.

In this study, 169,652 *C. oleifera* unigenes were annotated using seven public databases (NR, NT, Swiss-Prot, Pfam, GO, KOG, and KEGG), with 52.79% (89,569) successfully matched in at least one. While only 1.43% were annotated across all databases, Pfam, GO, and NT contributed the most annotations. Pfam, accounting for 58.95% of annotated unigenes, highlighted conserved protein domains potentially linked to metabolism, stress responses, and cell wall formation—processes essential during germination [35]. GO annotations (34.01%) offered insights into biological processes, cellular components, and molecular functions, suggesting roles in cell wall organization, hormone signaling, and metabolic regulation. These findings align with studies in Arabidopsis and other crops, where GO enrichment has identified key genes involved in early development and stress adaptation [36]. For example, by identifying genes related to hormone signaling, cell wall modification, and stress tolerance through GO annotation, researchers can select more stable and resilient germplasm, ultimately enhancing seedling propagation and supporting targeted breeding efforts for improved crop performance [37,38,39,40].

Annotations from the NT (31.42%) and Swiss-Prot (30.84%) databases enriched insights into evolutionary relationships and protein functions. Swiss-Prot, being manually curated, offers high-confidence functional annotations, especially for conserved proteins involved in developmental processes [41]. NT provides broader coverage, though with less specificity. These annotations enhance the completeness of the dataset and enable accurate functional predictions. Despite this, the NR and KOG annotation rates were relatively low (11.50% and 11.67%), likely due to limited genomic resources for *C. oleifera*, a non-model species [42]. Still, successful annotations from Pfam, GO, and KEGG compensated for this limitation. KEGG analysis revealed that 15.89% of the unigenes participated in key metabolic and signaling pathways, including carbohydrate metabolism, photosynthesis, and hormone signal transduction, which are vital for seed germination [43]. These functional insights establish a solid basis for downstream gene expression and pathway analysis. Future work will aim to identify key regulatory genes that influence germination and seedling vigor in *C. oleifera* cultivation. KOG annotation classified the unigenes into 25 categories, reflecting the complexity of the transcriptome. The category [O] “Translation, ribosomal structure and biogenesis” (~15%), indicates high protein synthesis during early development [44]. Category [R] “General function prediction only” (~11%) reflects many uncharacterized genes, typical of non-model species [45], while [T] “Signal transduction mechanisms” indicates active regulatory networks involved in growth and stress responses [46].

Other enriched categories—[C] “Energy production,” [U] “Intracellular trafficking,” [A] “RNA processing,” [G] “Carbohydrate metabolism,” and [E] “Amino acid metabolism” (each 5–7%)—highlight the dynamic metabolism and structural activities essential for seed development [47]. Less represented categories like [Y] “Nuclear structure” and [V] “Defense mechanisms” may indicate tissue-specific expression or lower activity levels. GO classification grouped 57,706 unigenes into 56 subcategories, mainly involving cellular and metabolic processes, reinforcing the high metabolic activity during germination. Abundant genes linked to cell components and molecular functions such as binding and catalytic activity suggest active biochemical processes in oil-rich tissues [9,48].

The number of differentially expressed genes (DEGs) increased progressively with gemination stage. Moreover, at different gemination stages, the number of upregulated DEGs exceeded that of downregulated DEGs, which was similar to the results of studies of *Coix lachryrma-jobi* L. [49] and *Euryale ferox* Salisb [50], indicating that *C. oleifera* mainly positively regulates seed germination, growth, and development through these upregulated DEGs.

Gene Ontology (GO) enrichment analysis of differentially expressed genes (DEGs) across germination stages relative to the control (G0) revealed significant functional shifts consistent with dynamic biological responses in *C. oleifera*. The enrichment patterns observed in cellular components, biological processes, and molecular function categories provide insights into the molecular mechanisms underpinning the plant’s adaptation and metabolic adjustments during the germination stages. Within the cellular component category, a pronounced enrichment of DEGs associated with cell wall-related structures—such as “cell wall,” “external encapsulating structure,” and “apoplast”—was observed, with gene counts increasing from G1 to G4. This trend suggests active remodeling of the cell wall architecture, a process critical for plant development, stress response, and intercellular communication [51,52]. The involvement of the “photosystem II oxygen evolving complex” also points to modulation of the photosynthetic apparatus, potentially reflecting changes in energy capture and photoprotection mechanisms in response to germination stage conditions [53].

In the biological process category, the enrichment of terms related to “cell wall organization or biogenesis” and metabolic pathways involving glucans, polysaccharides, and carbohydrates aligns with enhanced cell wall synthesis and carbohydrate metabolism during the germination stages. Such processes are essential for maintaining structural integrity and providing substrates for energy metabolism and signaling [54,55]. Particularly notable was the increasing enrichment of the oxidation-reduction process, which rose dramatically from 137 DEGs at G1 to 724 at G4. This highlights an elevated redox activity, likely reflecting the plant’s oxidative stress response and redox homeostasis adjustment to germination stage-induced stimuli [56].

Molecular function enrichment patterns corroborated these findings, with increasing numbers of DEGs associated with “structural constituent of cell wall,” “heme binding,” and notably “oxidoreductase activity.” The latter’s strong enrichment underscores the critical role of redox enzymes in detoxification, signaling, and metabolic regulation under stress or developmental shifts [57]. Additional functions such as “transferase activity,” “pectinesterase activity,” and “antioxidant activity” reveal a complex network of enzymatic processes facilitating cell wall modification, signal transduction, and reactive oxygen species scavenging, essential for maintaining cellular homeostasis and adaptive responses [58].

KEGG pathway enrichment analysis revealed that DEGs were significantly enriched in several key pathways closely associated with seed germination and early growth. The phenylpropanoid biosynthesis pathway (ko00940) showed marked enrichment, with DEGs increasing from 40 at G1 to 122 at G4. This pathway is central to the production of a wide range of secondary metabolites such as lignin, flavonoids, and other phenolics, which are essential for a plant’s structural integrity, defense, and adaptation to environmental stresses [59,60]. Key enzymes in this pathway, including PAL, which catalyzes the conversion of phenylalanine to trans-cinnamic acid, play a pivotal role in lignin synthesis, defense responses, photosynthesis, and pigment production [61,62]. Increased PAL expression has been linked to enhanced lignin content in *Capsicum chinense* and rice [63,64]. Similarly, 4CL, involved in regulating lignin metabolism and flavonoid biosynthesis, was also upregulated, correlating with higher lignin content in *Ginkgo biloba* [65]. The enhanced expression of these DEGs may support antioxidant defenses and cell wall reinforcement, which are crucial for seed germination.

Photosynthesis-related pathways, including photosynthesis–antenna proteins (ko00196) and photosynthesis (ko00195), were consistently enriched, with DEGs ranging from 16 to 41 across the stages. This suggests modulation of light-harvesting complexes and the photosynthetic electron transport chain, possibly reflecting adjustments in energy capture and conversion to optimize growth or stress tolerance [66,67]. Genes encoding *LHCB* and *LHCA* proteins, integral to Photosystem I (PSI) and Photosystem II (PSII), were upregulated, enhancing light capture and photosynthetic efficiency. These findings align with previous studies of *Festuca kryloviana* and *Brassica campestris*, where PSI/PSII and *LHCB* gene expression supported auto-trophic growth and seedling establishment [68,69]. Our physiological data also indicated continuous increases in chlorophyll content during germination, further supporting transcriptional upregulation of photosynthesis-related genes and validating this response at the physiological level.

The starch and sucrose metabolism pathway (ko00500) also showed progressive enrichment, indicating active carbohydrate metabolism during seed germination. Genes involved in this pathway, such as SUS, were upregulated to promote fructose accumulation, alleviate osmotic pressure, and support cellular energy demand [70]. Similarly, the plant hormone signal transduction pathway (ko04075) exhibited increased DEG counts, underscoring the importance of hormonal regulation in seed germination. Genes involved in the auxin (IAA), cytokinin (CTK), and cyclin pathways—including IAA, *CYCD3*, and *AUX1*—were significantly upregulated, contributing to cell division and elongation [71,72,73]. The increases in IAA and CTK content during germination stages further corroborated the transcriptomic findings, suggesting hormone-mediated coordination of seed development.

Hormonal signaling plays a pivotal role in regulating seed dormancy, germination, and early seedling development. In the present study, the expression levels of IAA, *AUX1*, *CYCD3*, *ABF3*, and *MYC2* genes increased progressively with the advancement of seed germination in *C. oleifera*. These transcriptional changes were consistent with the elevated contents of CTK and IAA, suggesting that hormonal genes function through a complex regulatory network to promote seed dormancy release and initiate germination. Previous studies have demonstrated that IAA genes are essential for seed coat rupture and rapid germination. In *Eucommia ulmoides* and *Rheum pumilum*, upregulation of IAA genes facilitated auxin accumulation, thus enhancing the seed’s ability to germinate and support subsequent developmental transitions [74,75]. Similarly, in the oil palm (*Elaeis guineensis*), *CYCD3* gene expression was found to be critical to embryo growth as a promoter of cell elongation and division [76]. In *Arabidopsis thaliana*, the *AUX1* gene plays a dual role in enhancing auxin influx and relieving abscisic acid (ABA)-mediated inhibition of germination [77]. Moreover, *ABF3*, a core component of the ABA signaling pathway, regulates circadian rhythm and seed dormancy by modulating stress-responsive gene expression [78]. Upregulation of *ABF* genes during seed germination has also been observed in apples (*Malus domestica*) and wheat (*Triticum aestivum*) [79,80]. *MYC2*, a key transcriptional regulator of jasmonic acid (JA) signaling, has been implicated in modulating root growth inhibition and defense responses [81,82]. The interplay among these hormones is finely regulated: auxin and gibberellin act synergistically to promote cell elongation, whereas ABA antagonizes gibberellin signaling to suppress germination. Collectively, the co-expression of *IAA*, *AUX1*, *CYCD3*, *ABF3*, and *MYC2* genes during *C. oleifera* seed germination provides strong evidence that these genes play pivotal roles in breaking dormancy and initiating developmental programs. These findings offer an integrated view of the hormonal regulatory networks involved in oil tea seed germination and lay the groundwork for future genetic or hormonal manipulation strategies to enhance germination efficiency in this economically important species.

Transcription factors, such as B3, MYB, bHLH, and NAC, were also upregulated during seed germination, highlighting their regulatory roles in seed development and the release from dormancy. B3 family transcription factors have been shown to regulate the biosynthesis of key hormones such as ABA and gibberellin biosynthesis (GA), thereby influencing seed development and dormancy control [83]. MYB transcription factors modulate gene expression involved in seed coat biosynthesis pathways, as demonstrated in *Arabidopsis thaliana* [84], further supporting their contribution to physical dormancy release. In rice, the OsNAC3 transcription factor regulates seed germination by directly influencing the expression of OsEXP4, which promotes embryo cell elongation, and by indirectly modulating OsEXP4 expression through ABA signaling [85]. Similarly, NAC genes in our study were upregulated during seed germination in *C. oleifera*, suggesting a conserved mechanism across species. Together, these transcription factors form a complex regulatory network that controls hormonal balance, cell elongation, and developmental transitions, reinforcing their essential roles in promoting seed germination and early seedling growth. These findings further support the overall conclusions of the study and offer new insights into the transcriptional control mechanisms of seed germination.

## 4. Materials and Methods

### 4.1. Study Site

The experiment was conducted at the Experimental Forest Farm of the Hunan Academy of Forestry, located in Yuhua District, Changsha City, Hunan Province, China (Latitude: 28°06′ N, Longitude: 113°01′ E). The site is characterized by a typical subtropical monsoon humid climate, with an average annual temperature of 17.2 °C, approximately 1762.64 h of sunshine per year, and an average annual precipitation of 1361.6 mm. The region experiences a frost-free period of around 290 days annually. The soil at the site is predominantly acidic red soil (Ultisols), with a well-developed and deep profile, providing favorable conditions for plant growth.

### 4.2. Plant Materials

‘Xianglin 27’ *C. oleifera* seeds were used as the experimental material. A seed germination experiment was conducted from 27 December 2018 to 21 January 2019. The seeds used in this study were collected from mature fruits of the *C. oleifera* cultivar ‘Xianglin 27’ in late October 2018 at the Experimental Forest Farm of the Hunan Academy of Forestry. The harvested fruits were air-dried in the shade, dehulled to extract seeds, and the seeds were sealed in airtight bags and stored at 4 °C for over two months. This cold storage period allowed partial release from physiological dormancy, which is commonly required for successful germination of *C. oleifera* seeds. While scarification is not typically necessary, cold stratification (e.g., sand storage during winter) is a standard practice for producing rootstocks used in grafting. In our case, storage under cool conditions was sufficient to initiate germination under laboratory conditions. On 26 December 2018, the seeds were surface sterilized by soaking in 0.3% potassium permanganate (KMnO_4_) solution for 10 min, rinsed thoroughly with deionized water, and then soaked in deionized water for 24 h. On 27 December 2018, the germination experiment was initiated. Sterilized sponge pads and filter paper were placed in petri dishes, followed by the addition of 50 mL of deionized water. Seeds were placed on the filter paper and incubated in a growth chamber set at 25 °C with 30% humidity, under a 12 h light/12 h dark photoperiod. Germination was defined as radicle emergence through the seed coat, and filter paper moisture was maintained by periodic spraying.

This study focused on analyzing physiological and transcriptomic changes during different developmental stages of *C. oleifera* seed germination to better understand the underlying mechanisms and provide a reference for selecting vigorous rootstocks in seedling propagation. Based on continuous observations, five germination stages were defined using visible morphological characteristics and developmental timing: G0 (Day 1), initial imbibition stage with no radicle emergence. G1 (Day 12), radicle just breaking through the seed coat. G2 (Day 17), radicle elongates to approximately 2 cm. G3 (Day 21), radicle elongates to approximately 5 cm. G4 (Day 26), full seedling development, with both shoot and cotyledon emergence (Figure 1). These stages were based on visible morphological features and referenced comparable approaches from previous studies [7,21,86,87] to ensure consistency and comparability across germination research. For each stage, representative seeds clearly exhibiting the defining characteristics were selected. After being thoroughly rinsed with distilled water, the seeds were immediately flash-frozen in liquid nitrogen (−80 °C) and stored for subsequent analysis.

### 4.3. Physiological and Biochemical Parameter Measurements

Phytohormones and key biochemical markers were quantified to assess the physiological status of the plants. The contents of indole-3-acetic acid (IAA) and cytokinin (CTK) were determined using a competitive enzyme-linked immunoassay (ELISA) method, following the protocols described by [88]. Frozen tissue samples (0.5 g) were homogenized in nine volumes of 0.9% physiological saline under ice bath conditions to prepare a 10% homogenate. The homogenate was centrifuged at 2500 rpm for 10 min, and the supernatant was collected for analysis. Each sample was measured in triplicate. The ELISA procedure included coating antibodies onto polystyrene plates, blocking nonspecific binding sites, incubation with samples or controls, substrate reaction with TMB, and optical density measurements at 450 nm, with concentrations calculated based on a standard curve. Antioxidant enzyme activities were assessed by extracting 0.5 g of frozen tissue with 5 mL of pre-chilled 50 mmol·L^−1^ phosphate buffer (pH 7.0). The mixture was ground on ice and centrifuged at 12,000 rpm for 10 min at 4 °C. The supernatant served as a crude enzyme extract for the detection of superoxide dismutase (SOD), peroxidase (POD), and catalase (CAT). SOD activity was measured using the nitroblue tetrazolium (NBT) method, CAT activity by UV spectrophotometry, and POD activity via the guaiacol assay, following established protocols [89,90]. Ascorbate peroxidase (APX) activity was determined by grinding 0.5 g of frozen tissue in liquid nitrogen with 5 mL of cold phosphate buffer (pH 7.8), followed by centrifugation at 12,000 rpm for 10 min at 4 °C. Enzyme activity was measured by absorbance at 290 nm [91]. Lipid peroxidation was quantified by measuring malondialdehyde (MDA) levels using the thiobarbituric acid (TBA) method. Frozen leaf samples (0.5 g) were ground in liquid nitrogen and homogenized in 10 mL of 10% trichloroacetic acid. The homogenate was centrifuged at 4000 rpm for 10 min, and 2 mL of the supernatant was reacted with 2 mL of 0.6% TBA solution. Absorbance readings were taken at 450, 532, and 600 nm [92]. Hydrogen peroxide (H_2_O_2_) content was assessed by homogenizing 0.1–0.5 g of tissue in 1 mL of acetone under ice bath conditions. After centrifugation at 12,000 rpm for 10 min at 4 °C, the supernatant absorbance was measured at 415 nm [93]. Photosynthetic pigments were extracted from 0.15 g of frozen leaves ground in liquid nitrogen with 80% acetone [94]. Following centrifugation, the supernatant absorbance was recorded at 649 nm and 665 nm. Chlorophyll a (Chl a), chlorophyll b (Chl b), and total chlorophyll (Chl a + b) concentrations were calculated. Soluble protein content was determined using the Coomassie Brilliant Blue G-250 method [95]. Total soluble sugar, sucrose, and starch contents were quantified using the anthrone-sulfuric acid and anthrone colorimetric methods, respectively [96].

### 4.4. RNA Extraction and Transcriptome Sequencing

Total RNA was extracted from pooled seed samples representing the five germination stages using TRIzol reagent (Invitrogen, Carlsbad, CA, USA), following the manufacturer’s protocol. RNA integrity and potential contamination were first assessed by agarose gel electrophoresis. Purity and integrity were further evaluated using a NanoPhotometer spectrophotometer (Thermo Scientific, Wilmington, DE, USA) and an Agilent 2100 Bioanalyzer (Agilent Technologies, Santa Clara, CA, USA). RNA sequencing libraries were constructed using the NEBNext^®^ Ultra™ RNA Library Prep Kit for Illumina^®^ (NEB, Ipswich, MA, USA). cDNA fragments of 250–300 bp were selected and purified using the AMPure XP system (Beckman Coulter, Beverly, CA, USA). Library quality was verified using the Agilent 2100 Bioanalyzer. High-throughput sequencing was performed on the Illumina platform by Novogene Co.; Ltd. (Beijing, China).

### 4.5. De Novo Assembly and Functional Annotation

Raw reads in FASTQ format were subjected to quality control using in-house Perl scripts. Reads containing adapter sequences, poly-N regions, or a high proportion of low-quality bases were removed to generate clean reads. The quality of the clean reads was evaluated based on Q20 and Q30 scores, GC content, and sequence duplication rates. De novo assembly of the clean reads was performed using Trinity (version 2.11.0), resulting in a comprehensive set of assembled transcripts. To minimize redundancy and group isoforms, Corset was used for hierarchical clustering, producing a non-redundant unigene set. Functional annotation of the assembled unigenes was conducted via sequence similarity searches against multiple public databases, including the NCBI non-redundant protein database (NR), Clusters of Orthologous Groups (COG), Swiss-Prot, Pfam, Kyoto Encyclopedia of Genes and Genomes (KEGG), and Gene Ontology (GO). Sequence alignment was carried out using BLASTX with an E-value threshold of <1 × 10^−5^ to assign putative gene functions and biological pathways.

### 4.6. Differentially Expressed Gene (DEG) Identification and Enrichment Analysis

To identify genes exhibiting significant changes in expression across germination stages, differential expression analysis was performed using the DESeq R package (version 1.10.1). Genes with an adjusted *p*-value (*p*-adj) < 0.05 and an absolute log2 fold change (|log2FC|) > 1 were considered significantly differentially expressed. Gene Ontology (GO) enrichment analysis of differentially expressed genes (DEGs) was performed using the GOseq R package, which accounts for gene length bias in the RNA-seq data. KEGG pathway enrichment analysis was performed using KOBAS (version 2.0.12), identifying significantly enriched metabolic and signaling pathways. These analyses provided key insights into the biological functions and regulatory mechanisms associated with seed germination and early seedling development in *C. oleifera*.

### 4.7. Quantitative Real-Time PCR (qRT-PCR) Analysis

To validate the transcriptome data, nine differentially expressed genes (DEGs) were selected for quantitative real-time PCR (qRT-PCR) analysis. Tubulin was used as the internal reference gene due to its stable expression across samples. Total RNA was reverse-transcribed into cDNA using a standard reverse transcription kit, and qRT-PCR was performed using gene-specific primers and SYBR Green chemistry. The relative expression levels of target genes were calculated using the 2^−ΔΔCt^ method. Each reaction was conducted in triplicate to ensure reproducibility and accuracy of the results.

## 5. Conclusions

In this study, we conducted a comprehensive analysis of the transcriptomic responses and physiological–biochemical parameters of *C. oleifera* seeds at five different stages of germination. Throughout the germination process, key phytohormones such as CTK and IAA, along with physiological indicators like chlorophyll content, soluble sugars, sucrose, protein, starch, and antioxidant enzyme activities, exhibited overall increases, reflecting significant molecular and metabolic changes. Notably, the G4 germination stage promoted the highest levels of hormones and starch content, while the G2 germination stage enhanced antioxidant activities and sugar accumulation, suggesting distinct regulatory pathways under different germination stages. A total of 169,652 unigenes were identified, with 89,569 successfully annotated, thereby enriching the transcriptomic resources for this species and providing a strong foundation for future molecular research. The number of differentially expressed genes (DEGs) increased progressively during germination, with a predominance of upregulated genes, indicating that *C. oleifera* primarily relies on gene activation to regulate seed germination, growth, and early development. Functional enrichment analyses revealed that DEGs were significantly involved in biological processes such as cell wall organization, carbohydrate metabolism, photosynthesis, and plant hormone signal transduction. Notably, the upregulation of cellulose synthase (CESA) genes suggests enhanced cellulose biosynthesis, crucial to maintaining cell wall structure and facilitating cell division and elongation. Key enzymes in the phenylpropanoid biosynthesis pathway—including PAL, 4CL, COMT, and POD—were also upregulated, indicating increased lignin synthesis that reinforces cell wall strength and structural integrity. Photosynthesis-related genes (*LHCB*, *LHCA*, *psa*, *psb*) were upregulated, enhancing photosynthetic capacity to provide the energy necessary for seed germination and seedling growth. Hormone-related genes (*IAA*, *AUX1*, *ABF*, *CYCD3*, *MYC2*) exhibited increased expression, reflecting a complex hormonal regulatory network that breaks seed dormancy and promotes embryonic development and cell proliferation. Additionally, multiple transcription factors (B3, bHLH, MYB, NAC) and other functional genes (*SUS*, *ANR*, *CHS*, *DFR*) showed progressive upregulation during germination, suggesting their roles in gene regulation, cell wall remodeling, and stress responses. Collectively, this study identified numerous candidate genes associated with *C. oleifera* seed germination and early development, providing valuable insights into the underlying molecular mechanisms and offering potential targets to improve breeding and cultivation strategies. Future research can further explore the interactions among these indicators and genes, and their adaptive changes under varying environmental conditions, to enhance the understanding of how *C. oleifera* can optimize seedling growth under stress and improve agricultural practices for this species.

## Figures and Tables

**Figure 1 plants-14-02283-f001:**
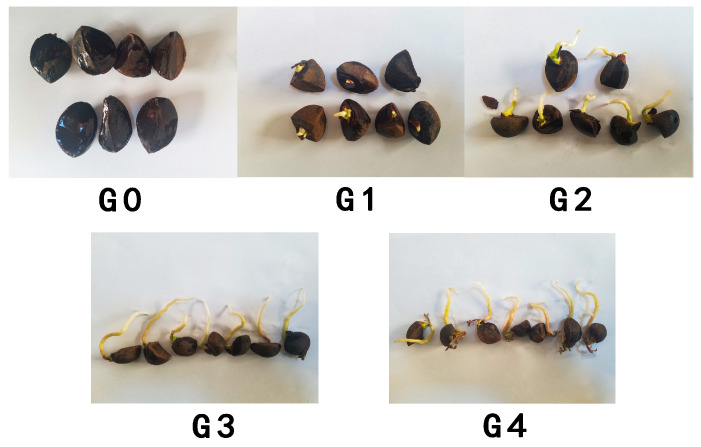
Morphological characteristics of *C. oleifera* seeds at five germination stages (G0–G4). G0 (Day 1), initial imbibition stage with no radicle emergence. G1 (Day 12), radicle just breaking through the seed coat. G2 (Day 17), radicle elongates to approximately 2 cm. G3 (Day 21), radicle elongates to approximately 5 cm. G4 (Day 26), full seedling development, with both shoot and cotyledon emergence.

**Table 1 plants-14-02283-t001:** Changes in phytohormone levels, photosynthetic pigments, carbohydrates, protein content, and antioxidant enzyme activities across different germination stages of *C. oleifera* seeds.

Germination Stage	CTK (µg/L)	IAA (µg/L)	Chl a (mg/g FW)	Chl b (mg/g FW)	Chl a + Chl b (mg/g FW)	Soluble Sugar (mg/g FW)	Sucrose (mg/g FW)	Protein (mg/g FW)	Starch (mg/mg prot)	SOD (U/mg prot)	POD (ΔOD_470_/min/mg prot)	APX (μmol/min/mg prot)	CAT (μmoL/min/mg prot)	MDA (nmol/mg prot)	H_2_O_2_ (μmol/mg prot)
G0	4.14 ± 0.11 c	4.65 ± 0.17 d	0.002 ± 0.000 a	0.005 ± 0.003 a	0.007 ± 0.003 a	61.86 ± 1.16 c	95.79 ± 1.39 e	4.14 ± 0.06 d	1.82 ± 0.03 e	3.84 ± 0.02 e	0.25 ± 0.00 e	0.12 ± 0.00 c	0.17 ± 0.02 d	0.16 ± 0.011 c	0.06 ± 0.00 e
G1	4.65 ± 0.05 b	5.08 ± 0.17 d	0.002 ± 0.000 a	0.009 ± 0.002 a	0.011 ± 0.002 a	64.91 ± 0.97 c	117.03 ± 1.14 c	10.78 ± 0.04 a	1.93 ± 0.03 d	10.24 ± 0.03 b	0.34 ± 0.01 d	0.32 ± 0.01 a	0.84 ± 0.01 c	0.31 ± 0.01 b	1.17 ± 0.01 a
G2	4.69 ± 0.03 b	6.01 ± 0.17 c	0.003 ± 0.002 a	0.011 ± 0.003 a	0.014 ± 0.005 a	75.73 ± 1.18 a	130.43 ± 0.67 a	8.33 ± 0.08 b	2.50 ± 0.01 b	27.48 ± 0.07 a	1.95 ± 0.02 a	0.32 ± 0.02 a	1.36 ± 0.02 b	0.71 ± 0.07 a	0.83 ± 0.01 b
G3	5.32 ± 0.19 a	6.95 ± 0.22 b	0.004 ± 0.000 a	0.011 ± 0.001 a	0.015 ± 0.001 a	72.02 ± 1.14 b	121.46 ± 0.72 b	6.38 ± 0.11 c	2.05 ± 0.02 c	9.31 ± 0.14 c	1.74 ± 0.02 b	0.13 ± 0.00 c	10.72 ± 0.01 a	0.81 ± 0.01 a	0.61 ± 0.01 c
G4	5.48 ± 0.10 a	8.12 ± 0.18 a	0.005 ± 0.003 a	0.017 ± 0.008 a	0.022 ± 0.011 a	62.37 ± 0.38 c	107.79 ± 0.65 d	4.32 ± 0.06 d	4.22 ± 0.05 a	5.48 ± 0.10 d	0.50 ± 0.01 c	0.18 ± 0.00 b	1.17 ± 0.18 b	0.22 ± 0.00 bc	0.40 ± 0.01 d

Note: Values are presented as means ± standard deviations (n = 3). Different lowercase letters within the same row indicate significant differences among germination stages (*p* < 0.05), as determined by one-way ANOVA followed by Tukey’s HSD post hoc test. FW: fresh weight.

**Table 2 plants-14-02283-t002:** Analysis of differentially expressed genes (DEGs).

Compare	All DEGs	Up DEGs	Down DEGs
G1 vs. G0	1708	1474	234
G2 vs. G0	6505	5636	869
G3 vs. G0	8206	7045	1161
G4 vs. G0	10,250	8221	2029

## Data Availability

The data will be provided by the authors upon reasonable request.

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
