# Peer review of "Stage-Specific Transcriptomic Insights into Seed Germination and Early Development in Camellia oleifera Abel."

_plants, 2025, doi:10.3390/plants14152283_

Round 1
Reviewer 1 Report
Comments and Suggestions for Authors
Dear Authors,
All the corrections are in pdf attached as comments.
kind regards

Author Response
Thank you very much for your valuable comments and constructive feedback on our manuscript. Your insights were instrumental in improving the clarity and scientific quality of our work. We truly appreciate your time and contribution to the review process. Below, we provide point-by-point responses to your comments. Our responses are underlined, followed by detailed explanations and the corresponding revisions made to the manuscript. The revised text is presented in italics and has been directly excerpted from the updated manuscript.
- L17. The abstract should be a total of about 200 words maximum - from "Instructions for Authors"
Re: Thank you for your valuable feedback regarding the abstract. In accordance with the Instructions for Authors and in response to another reviewer’s comments, we have revised the abstract. The updated abstract is as follows: “Seed germination is a critical phase in the plant lifecycle of Camellia oleifera (oil tea), directly influencing seedling establishment and crop reproduction. In this study, we examined transcriptomic and physiological changes across five defined germination stages (G0–G4), from radicle dormancy to cotyledon emergence. Using RNA sequencing (RNA-seq), we assembled 169,652 unigenes and identified differentially expressed genes (DEGs) at each stage compared to G0, increasing from 1,708 in G1 to 10,250 in G4. Functional enrichment analysis revealed upregulation of genes associated with cell wall organization, glucan metabolism, and photosystem II assembly. Key genes involved in cell wall remodeling, including cellulose synthase (CESA), phenylalanine ammonia-lyase (PAL), 4-coumarate-CoA ligase (4CL), caffeoyl-CoA O-methyltransferase (COMT), and peroxidase (POD), showed progressive activation during germination. Kyoto Encyclopedia of Genes and Genomes (KEGG) pathway analysis revealed dynamic regulation of phenylpropanoid and flavonoid biosynthesis, photosynthesis, carbohydrate metabolism, and hormone signaling pathways. Transcription factors such as indole-3-acetic acid (IAA), ABA-responsive element binding factor (ABF), and basic helix–loop–helix (bHLH) were upregulated, suggesting hormone-mediated regulation of dormancy release and seedling development. Physiologically, cytokinin (CTK) and IAA levels peaked at G4, antioxidant enzyme activities were highest at G2, and starch content increased toward later stages. These findings provide new insights into the molecular mechanisms underlying seed germination in C. oleifera and identify candidate genes relevant to rootstock breeding and nursery propagation.”.
- L21“fivedistinct germination stages”
Re: The Abstract has been revised. Please see the revision.
- L25
Re: The Abstract has been revised. Please see the revision.
- L30 PAL, 4CL, COMT, and POD. Abreviations should be explained in Abstract, but I leave the decission to the Editor.
Re: Thank you for the suggestion. We have added the full names of these genes in the Abstract accordingly. In addition, we have added the full names of other abbreviations throughout the manuscript at their first occurrence.
- L107 Line Please describe more about what type of seed Camellia oleifera has. Is it recalcitrant, needing stratification and scarification to germinate? Is the embryo in deep physiological dormancy, or is it undeveloped?I think that such information is crucial in planning any experiments.
Re: Thank you for your valuable comment and suggestion. We have added additional information to the Methods section (4.2 Plant Materials) of the revised manuscript to clarify the dormancy characteristics of Camellia oleifera seeds. The revised text reads: “Based on continuous observations, five germination stages were defined using visible morphological characteristics and developmental timing: G0 (Day 1), initial imbibition stage with no radicle emergence. G1 (Day 12), radicle just breaking through the seed coat. G2 (Day 17), radicle elongates to approximately 2 cm. G3 (Day 21), radicle elongates to approximately 5 cm. G4 (Day 26), full seedling development, with both shoot and cotyledon emergence (Figure 1). These stages were based on visible morphological features and referenced comparable approaches from previous studies [7, 21, 86-87] to ensure consistency and comparability across germination research.”.
6.L136 Line treatments? you didn't treated the seeds, maybe you ment "stages". A "treatment" is a specific condition or intervention applied to experimental units, but according to m&m you didn't treated the seeds you just selected the stages of germination and place in LN. Is this right?
Re: Thank you for your helpful observation. We have revised the manuscript and replaced the word “treatment” with “germination stage” in the revision.
- L141 CTK explain abbreviation
Re: Thank you for your comment. We have now explained the abbreviation cytokinin (CTK) at its first occurrence in the revised manuscript.
- L144 grammar, please construct sentences correctly.
Re: We have revised the sentence to read “No significant differences were found in photosynthetic pigments….” in the revision.
- L145 Chl a, Chl b, explain abbreviation
Re: Thank you for your comment. We have added the full names for the abbreviations chlorophyll a (Chl a) and chlorophyll b (Chl b) at their first occurrence in the revised manuscript.
- L153 SOD and POD explain abbreviation
Re: Thank you for your comment. We have added the full names for the abbreviations superoxide dismutase (SOD) and peroxidase (POD) at their first occurrence in the revised manuscript.
- L164 Clean readsplease add one sentence explaining what was the plant matherial that you examined for transcriptome sequencing
Re: Thank you for your comment. We have added a sentence in the revised manuscript to clarify the plant material used for transcriptome sequencing. It reads that, “Germinated seeds of Camellia oleifera at different developmental stages were used for RNA extraction prior to sequencing.”.
- L435 “Studies on Arabidopsis and other crop species have demonstrated that GO annotations can reveal key genes involved in seedling development and stress responses, which are highly relevant for improving agricultural practices in C. oleifera [33].” Can you provide example how this finding will improve agricultural practices in C. oleifera?
Re: Thank you for your thoughtful comment. Our intention was to convey that GO annotation enables the identification of key genes potentially involved in seed germination and early development in Camellia oleifera. This understanding provides valuable insights into the molecular mechanisms underlying seedling establishment. For example, identifying genes related to hormone signaling, cell wall modification, or stress tolerance can inform the selection of stable germplasm and improve seedling propagation strategies, thereby supporting more efficient and targeted C. oleifera breeding and nursery practices. In response, we have added examples in the revised manuscript. It now reads “For example, by identifying genes related to hormone signaling, cell wall modification, and stress tolerance through GO annotation, researchers can select more stable and resilient germplasm, ultimately enhancing seedling propagation and supporting targeted breeding efforts for improved crop performance [37-40].”.
L603 4.1 Study SiteCan you explain why you describe the study site? Did you collect seeds from Camelia plants growing there? When did you collect those seeds? How did you handle them later?
Re: Thank you for your questions. Yes, we collected seeds from Camellia oleifera plants growing at the National Camellia oleifera Germplasm Collection and Conservation Base of the Hunan Academy of Forestry Sciences. We have included additional information in the manuscript. The added paragraph reads as follows: “Based on continuous observations, five germination stages were defined using visible morphological characteristics and developmental timing: G0 (Day 1), initial imbibition stage with no radicle emergence. G1 (Day 12), radicle just breaking through the seed coat. G2 (Day 17), radicle elongates to approximately 2 cm. G3 (Day 21), radicle elongates to approximately 5 cm. G4 (Day 26), full seedling development, with both shoot and cotyledon emergence (Figure 1). These stages were based on visible morphological features and referenced comparable approaches from previous studies [7, 21, 86-87] to ensure consistency and comparability across germination research.”.
- L613 ‘Xianglin 27’ oleifera seeds were used as the experimental material and a seed germination experiment was conducted from December 27, 2018, to January 21, 2019.”where did you buy the seeds? or did you collect them? if so when? did they need startification.
Re: Thank you for your question. The Camellia oleifera seeds used in our experiment were not purchased but were collected in late October 2018 from mature fruits of the cultivar ‘Xianglin 27’ at the Experimental Forest Farm of the Hunan Academy of Forestry. For the purposes of this experiment, we did not apply formal stratification treatment. After harvest, the fruits were air-dried in the shade, dehulled to extract the seeds, and the seeds were sealed in airtight bags and stored at 4 °C. On December 26, 2018, the seeds were disinfected in 0.3% potassium permanganate (KMnO₄) solution for 10 minutes, rinsed thoroughly with deionized water, and soaked for 24 hours. The germination experiment was initiated on December 27, 2018.
- L614-615 “Five distinct germination stages were defined based on visible morphological characteristics: G0, radicle not yet emerged from the seed coat;”
How was the germination procedure performed? Was it on filter paper? Or in a substrate? And if so, what substrate and at what temperature? Under what light conditions? For how long? In which laboratory? Please describe the germination procedure in detail, including the duration of each phase. What determined (e.g., root length) the germinating embryo to be assigned to the next stage? Can you cite literature describing the methodology for this stage of germination and the naming of the individual stages?
Re: Thank you for your detailed comment. We have revised the manuscript to include a more complete description of the germination procedure and the criteria used to define the germination stages. We have revised the manuscript in the Methods section (4.2 Plant Materials) as follows: “The seeds used in this study were collected from mature fruits of the Camellia oleifera cultivar 'Xianglin 27' in late October 2018 at the Experimental Forest Farm of the Hunan Academy of Forestry. The harvested fruits were air-dried in the shade, dehulled to extract seeds, and the seeds were sealed in airtight bags and stored at 4 °C for over two months. This cold storage period allowed partial release from physiological dormancy, which is commonly required for successful germination of C. oleifera seeds. While scarification is not typically necessary, cold stratification (e.g., sand storage during winter) is a standard practice for producing rootstocks used in grafting. In our case, storage under cool conditions was sufficient to initiate germination under laboratory conditions. On December 26, 2018, the seeds were surface sterilized by soaking in 0.3% potassium permanganate (KMnO₄) solution for 10 minutes, rinsed thoroughly with deionized water, and then soaked in deionized water for 24 hours. On December 27, 2018, the germination experiment was initiated. Sterilized sponge pads and filter paper were placed in Petri dishes, followed by the addition of 50 mL of deionized water. Seeds were placed on the filter paper and incubated in a growth chamber set at 25 °C with 30% humidity, under a 12 h light / 12 h dark photoperiod. Germination was defined as radicle emergence through the seed coat, and filter paper moisture was maintained by periodic spraying.
This study focused on analyzing physiological and transcriptomic changes during different developmental stages of C. oleifera seed germination to better understand the underlying mechanisms and provide a reference for selecting vigorous rootstocks in seedling propagation. Based on continuous observations, five germination stages were defined using visible morphological characteristics and developmental timing: G0 (Day 1), initial imbibition stage with no radicle emergence. G1 (Day 12), radicle just breaking through the seed coat. G2 (Day 17), radicle elongates to approximately 2 cm. G3 (Day 21), radicle elongates to approximately 5 cm. G4 (Day 26), full seedling development, with both shoot and cotyledon emergence (Figure 1). These stages were based on visible morphological features and referenced comparable approaches from previous studies [7, 21, 86-87] to ensure consistency and comparability across germination research.”
- “concerning chapters 4.3-4.7”I am not sure if description of the methods used in measurements of physiological parameters can be presented so briefly, referring to the kit name and manufacturer's recommendations. This is the first time I've encountered such a description in scientific publication. In my opinion, such a superficial presentation of the method is insufficient, and it soould be far more detailed, but I'll leave the decision to the Editor.
Re: Thank you for your insightful comment regarding Sections 4.3–4.7. We appreciate your concern about the level of methodological detail provided for the physiological measurements. In the original submission, we referenced the use of standardized commercial kits to streamline the text. However, in response to your suggestion, we have now expanded the Methods section to include detailed descriptions of the experimental procedures used for the quantification of indole-3-acetic acid (IAA), cytokinin (CTK), antioxidant enzyme activities (SOD, POD, CAT, APX), malondialdehyde (MDA), hydrogen peroxide (H₂O₂), photosynthetic pigments (Chl a, Chl b), protein, sugar, and starch content. The revised text includes information on tissue preparation, buffer compositions, extraction protocols, centrifugation parameters, absorbance wavelengths, and the calculation methods applied. Please see the revised 4.3. Physiological and Biochemical Parameter Measurements: “The contents of indole-3-acetic acid (IAA) and cytokinin (CTK) were determined using a competitive enzyme-linked immunoassay (ELISA) method, following the protocols described by [88]. Frozen tissue samples (0.5 g) were homogenized in nine volumes of 0.9% physiological saline under ice bath conditions to prepare a 10% homogenate. The homogenate was centrifuged at 2,500 rpm for 10 minutes, and the supernatant was collected for analysis. Each sample was measured in triplicate. The ELISA procedure included coating of antibodies onto polystyrene plates, blocking nonspecific binding sites, incubation with samples or controls, substrate reaction with TMB, and optical density measurement at 450 nm, with concentrations calculated based on a standard curve. Antioxidant enzyme activities were assessed by extracting 0.5 g of frozen tissue with 5 mL of pre-chilled 50 mmol·L⁻¹ phosphate buffer (pH 7.0). The mixture was ground on ice and centrifuged at 12,000 rpm for 10 minutes at 4 °C. The supernatant served as a crude enzyme extract for determination of superoxide dismutase (SOD), peroxidase (POD), catalase (CAT). SOD activity was measured using the nitroblue tetrazolium (NBT) method, CAT activity by UV spectrophotometry, and POD activity via the guaiacol assay, following established protocols [89-90]. Ascorbate peroxidase (APX) activity was determined by grinding 0.5 g of frozen tissue in liquid nitrogen with 5 mL of cold phosphate buffer (pH 7.8), followed by centrifugation at 12,000 rpm for 10 minutes at 4 °C. Enzyme activity was measured by absorbance at 290 nm [91]. Lipid peroxidation was quantified by measuring malondialdehyde (MDA) levels using the thiobarbituric acid (TBA) method. Frozen leaf samples (0.5 g) were ground in liquid nitrogen and homogenized in 10 mL of 10% trichloroacetic acid. The homogenate was centrifuged at 4,000 rpm for 10 minutes, and 2 mL of the supernatant was reacted with 2 mL of 0.6% TBA solution. Absorbance readings were taken at 450, 532, and 600 nm [92]. Hydrogen peroxide (H₂O₂) content was assessed by homogenizing 0.1–0.5 g of tissue in 1 mL of acetone under ice bath conditions. After centrifugation at 12,000 rpm for 10 minutes at 4 °C, the supernatant absorbance was measured at 415 nm [93]. Photosynthetic pigments were extracted from 0.15 g of frozen leaves ground in liquid nitrogen with 80% acetone [94]. Following centrifugation, the supernatant absorbance was recorded at 649 nm and 665 nm. Chlorophyll a (Chl a), chlorophyll b (Chl b), and total chlorophyll (Chl a + b) concentrations were calculated. Soluble protein content was determined using the Coomassie Brilliant Blue G-250 method [95]. Total soluble sugar, sucrose, and starch contents were quantified using the anthrone-sulfuric acid and anthrone colorimetric methods, respectively [96].”.

Reviewer 2 Report
Comments and Suggestions for Authors
The manuscript deals with transcriptomic and biochemical characterization of Camellia oleifera seeds germination. It is novel study and produced many new data, however it need to be thoroughly rewritten to present them properly. There are many pittfals and imbalancies in the current manuscript.
Namely: Abstract is too long and should be rewritten to clearly translate the major findinds. It is currently convulated and complex.
Introduction part: skip already well known facts about RNAseq and transcriptomic analysis. Focus on the seed germination comparatively at relevant species, pick the well described pathways.
Results part: Do not speak about "treatments" as these are develomental stages ! You have not done any specific treatment, you simply let the seeds to germinate. Properly present all biochemical data in figures, tables - this includes hormones and other measured substances. These are important and should be later cross referred to transcriptomic data.
Many of the RNAseq tables and outcomes should be moved into supplementary material as these are not major findings, solely supportive indicators of quality. Focus on DEGs as identified between respective stages and describe them properly. Pick the most signifficant biochemical pathways and go more into details in term of expressed genes. If the species is grown for oil than it means that seeds have a oil storage, which is used during germination.
In this context, is the pre-germination seed development already studied in C. oleifera ?
Author Response
The manuscript deals with transcriptomic and biochemical characterization of Camellia oleifera seeds germination. It is novel study and produced many new data, however it need to be thoroughly rewritten to present them properly. There are many pittfals and imbalancies in the current manuscript.
Re: Thank you very much for taking the time to review our manuscript and for your constructive comments and suggestions. We appreciate your recognition of the novelty and the richness of the data in our study on the transcriptomic and biochemical characterization of Camellia oleifera seed germination. We acknowledge that the original version of the manuscript had shortcomings in presentation, including imbalances and unclear sections. In response, we have thoroughly revised the manuscript to enhance its clarity, structure, and scientific rigor. Below, we provide a point-by-point response to your comments. Our replies are underlined, followed by detailed explanations and descriptions of the corresponding revisions. The revised text is presented in italics and has been directly excerpted from the updated manuscript.
- Abstract is too long and should be rewritten to clearly translate the major findinds. It is currently convulated and complex.
Re: Thank you for your valuable feedback regarding the abstract. In response, we have substantially revised it to improve conciseness, clarity, and accessibility. The revised abstract presents the research background, objectives, key methodologies, major findings, and significance of the study in a clearer, more concise, and accessible way. The updated abstract is as follows: “Seed germination is a critical phase in the plant lifecycle of Camellia oleifera (oil tea), directly influencing seedling establishment and crop reproduction. In this study, we examined transcriptomic and physiological changes across five defined germination stages (G0–G4), from radicle dormancy to cotyledon emergence. Using RNA sequencing (RNA-seq), we assembled 169,652 unigenes and identified differentially expressed genes (DEGs) at each stage compared to G0, increasing from 1,708 in G1 to 10,250 in G4. Functional enrichment analysis revealed upregulation of genes associated with cell wall organization, glucan metabolism, and photosystem II assembly. Key genes involved in cell wall remodeling, including cellulose synthase (CESA), phenylalanine ammonia-lyase (PAL), 4-coumarate-CoA ligase (4CL), caffeoyl-CoA O-methyltransferase (COMT), and peroxidase (POD), showed progressive activation during germination. Kyoto Encyclopedia of Genes and Genomes (KEGG) pathway analysis revealed dynamic regulation of phenylpropanoid and flavonoid biosynthesis, photosynthesis, carbohydrate metabolism, and hormone signaling pathways. Transcription factors such as indole-3-acetic acid (IAA), ABA-responsive element binding factor (ABF), and basic helix–loop–helix (bHLH) were upregulated, suggesting hormone-mediated regulation of dormancy release and seedling development. Physiologically, cytokinin (CTK) and IAA levels peaked at G4, antioxidant enzyme activities were highest at G2, and starch content increased toward later stages. These findings provide new insights into the molecular mechanisms underlying seed germination in C. oleifera and identify candidate genes relevant to rootstock breeding and nursery propagation.”.
- Introduction part: skipalready well known facts about RNAseq and transcriptomic analysis. Focus on the seed germination comparatively at relevant species, pick the well described pathways.
Re: Thank you for your valuable feedback. We have revised the Introduction accordingly by removing general background information on RNA-seq and transcriptomic analysis. Instead, we have focused more specifically on seed germination processes, highlighting key molecular pathways and gene functions identified in Camellia oleifera and closely related species. This revision emphasizes relevant studies and well-characterized pathways involved in seed dormancy release, cell wall remodeling, hormone signaling, and stress responses during germination, providing a clearer and more focused context for our research. Please see the updated first paragraph of Introduction.
- Results part: Do not speak about "treatments" as these are develomental stages ! You have not done any specific treatment, you simply let the seeds to germinate. Properly present all biochemical data in figures, tables - this includes hormones and other measured substances. These are important and should be later cross referred to transcriptomic data.
Re: Thank you very much for your constructive feedback. In response, we have carefully revised the Results and Discussion sections to replace all references to "treatment" or "treatments" with "germination stage" or "stages," as appropriate. Additionally, all biochemical data have been clearly presented in figures and tables.
- Many of the RNAseq tables and outcomes should be moved into supplementary material as these are not major findings, solely supportive indicators of quality. Focus on DEGs as identified between respective stages and describe them properly. Pick the most signifficant biochemical pathways and go more into details in term of expressed genes.
Re: Thank you for your valuable suggestion. In response, we have moved several RNA-seq quality control tables and supporting data, including assembly statistics and basic annotation summaries—to the supplementary materials (Tables and Figures), as these serve as indicators of data quality rather than core findings. We have revised the main text to focus more explicitly on the differentially expressed genes (DEGs) identified between germination stages. In particular, we have expanded our analysis and discussion of key biochemical pathways, such as hormone signaling, phenylpropanoid biosynthesis, and cell wall modification. For these pathways, we have now provided more detailed descriptions of the significantly expressed genes and their potential roles during seed germination in the Discussion sections (3.2, 3.3, and 3.4). Please see the revised Discussion section.
- If the species is grown for oil than it means that seeds have a oil storage, which is used during germination. In this context, is the pre-germination seed development already studied in C. oleifera ?
Re: Thank you for your insightful comment. Yes, Camellia oleifera seeds are indeed rich in oil, primarily composed of unsaturated fatty acids, and this oil serves as a vital energy source during germination and early seedling development. However, while the processes of oil biosynthesis and accumulation during seed maturation have been studied, the molecular and physiological mechanisms of oil mobilization during seed germination in C. oleifera remain largely unexplored. Most existing research has focused on lipid composition and oil content for breeding and commercial purposes rather than on metabolic utilization during germination. Our study addresses this knowledge gap by combining transcriptomic and physiological data to investigate the regulation of stored lipid mobilization, including potential changes in lipid metabolism-related gene expression. These findings provide new insights into the energy dynamics supporting germination and seedling establishment in C. oleifera. We have added an additional paragraph to the Introduction, which states: “C. oleifera is cultivated primarily for its oil-rich seeds, which accumulate substantial quantities of triacylglycerols during seed development. These lipids serve as the main energy source during germination and are essential for fueling early seedling growth. While the biosynthesis and composition of seed oils in C. oleifera have been relatively well characterized [9-12], the mobilization and metabolic regulation of these reserves during germination remain poorly understood. Elucidating how lipid reserves are utilized during germination is crucial for understanding the energy metabolism underlying seedling establishment. This knowledge may further contribute to improving rootstock vigor and optimizing oilseed propagation strategies, both of which are central to the sustainable development of the C. oleifera industry.”.

Round 2
Reviewer 1 Report
Comments and Suggestions for Authors Dear Authors,The manuscript has been appropriately revised.
The work is now suitable for publication.
Reviewer 2 Report
Comments and Suggestions for Authors
I am pleased with the revised version.